# LLM-in-the-loop: Leveraging Large Language Model for Thematic Analysis

**Shih-Chieh Dai[1]**       **Aiping Xiong[2]**       **Lun-Wei Ku[3]**

[1] University of Texas at Austin       [2]Pennsylvania State University       [3]Academia Sinica

[1]sjdai@utexas.edu       [2]axx29@psu.edu       [3]lwku@iis.sinica.edu.tw

## Abstract

Thematic analysis (TA) has been widely used for analyzing qualitative data in many disciplines and fields. To ensure reliable analysis, the same piece of data is typically assigned to at least two human coders. Moreover, to produce meaningful and useful analysis, human coders develop and deepen their data interpretation and coding over multiple iterations, making TA labor-intensive and time-consuming. Recently the emerging field of large language models (LLMs) research has shown that LLMs have the potential replicate human-like behavior in various tasks: in particular, LLMs outperform crowd workers on text-annotation tasks, suggesting an opportunity to leverage LLMs on TA. We propose a human–LLM collaboration framework (i.e., LLM-in-the-loop) to conduct TA with in-context learning (ICL). This framework provides the prompt to frame discussions with a LLM (e.g., GPT-3.5) to generate the final codebook for TA. We demonstrate the utility of this framework using survey datasets on the aspects of the music listening experience and the usage of a password manager. Results of the two case studies show that the proposed framework yields similar coding quality to that of human coders but reduces TA's labor and time demands. [1]

## 1 Introduction

Braun and Clarke (2006) propose thematic analysis (TA) to identify themes that represent qualitative data (e.g. free-text response). TA, which relies on at least two human coders with relevant expertise to run the whole process, is widely used in qualitative research (QR). However, TA is labor-intensive and time-consuming. For instance, for an in-depth understanding of the data, coders require multiple rounds of discussion to resolve ambiguities and achieve consensus.

Large language models (LLMs) have advanced artificial intelligence (AI) tremendously recently (Brown et al., 2020; Chowdhery et al., 2022; Scao et al., 2022; Touvron et al., 2023). Prompting techniques such as in-context learning (ICL), chain-of-thought, and reasoning and action elicit decent results on various natural language processing (NLP) tasks (Wei et al., 2022; Kojima et al., 2022; Suzgun et al., 2022; Yao et al., 2023). LLM-based applications continue to multiply, including Copilot for programming,[2] LaMDA for dialogue generation (Thoppilan et al., 2022), and ChatGPT.[3] Indeed, several studies show that LLMs outperform human annotator performance on crowdsourcing tasks (Gilardi et al., 2023; Chiang and Lee, 2023; Ziems et al., 2023), which indicates a potential opportunity to leverage LLMs for TA. Thus, in this work, we address the research question: can NLP techniques (i.e., LLMs) enhance the efficiency of TA?

We loop ChatGPT to act as a machine coder (MC) and work with a human coder (HC) on TA. Following Braun and Clarke (2006), we leverage ICL to design the prompt for each step. The HC first becomes familiar with the data, after which the MC extracts the initial codes based on the research question and free-text responses. Next, the MC groups the similar initial codes into representative codes representing the responses and which answer the research question. We design a discussion prompt for the MC and HC to use to refine the codes. The discussion ends once the HC and MC come to agreement, after which the codebook is generated. Following the TA procedure, the HC and MC utilize the codebook to code the responses and calculate the inner-annotator agreement (IAA) to measure the quality of the work. See Section 3 for detailed information.

We study the effectiveness of this human–LLM

---

[2]https://github.com/features/copilot
[3]https://openai.com/blog/chatgpt

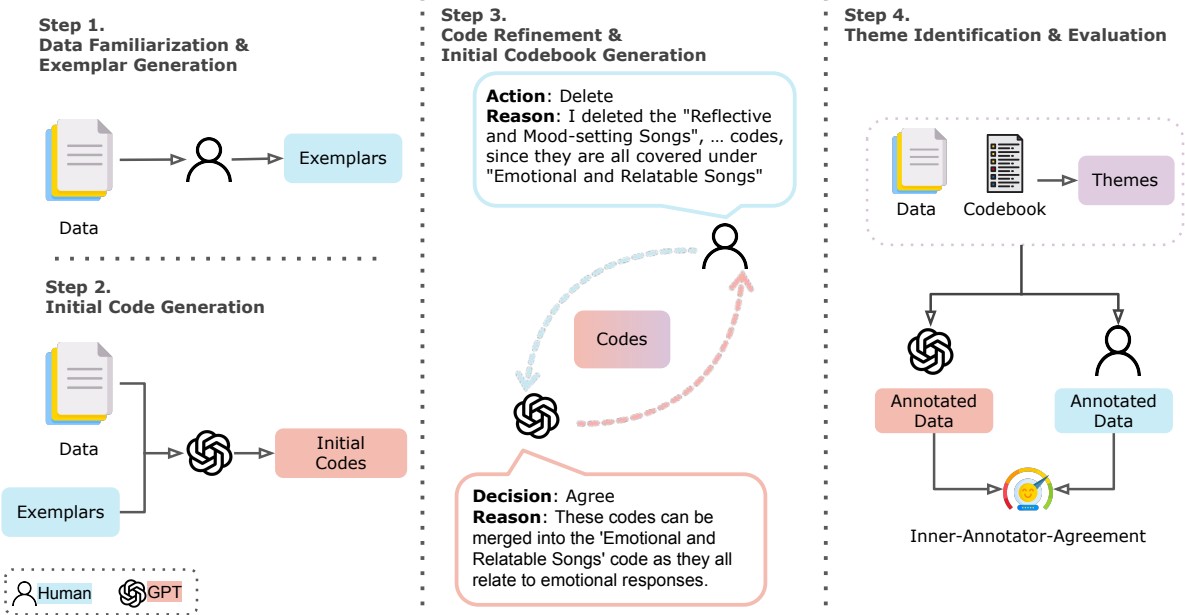

Figure 1: Workflow of the LLM-in-the-Loop Framework.

collaboration framework by adopting the framework on two cases: Music Shuffle (MS), a survey measuring aspects of music listening experiences (Sanfilippo et al., 2020) and Password Manager (PM), a survey on password manager usage (Mayer et al., 2022). Considering the LLM input size limitations, we randomly divided the responses of PM into two separate pools and applied one pool for codebook generation. Subsequently, we sampled responses from both pools for final coding and IAA calculation. The results suggest that the proposed LLM-in-the-loop framework yields work quality comparable to two HCs.

The contributions of this work are as follows: 1) we propose a human–LLM collaboration framework for TA; 2) we design a loop to facilitate discussions between a human coder and a LLM; 3) we propose a solution for long-text qualitative data when using a LLM for TA.

## 2 Background and Related Work

Given the substantial amount of free-text responses, it seems obvious that NLP techniques can be potential solutions to facilitate TA. Existing work tends to rely on topic modeling to extract key topics that represent the data (Leeson et al., 2019; Jelodar et al., 2020). Another approach is clustering: for instance, Oyebode et al. (2021) extract phrases, calculate their sentiment scores, and group the phrases into themes based on their sentiment polarity. Guetterman et al. (2018) cluster phrases by their Wu–Palmer similarity scores (Wu and Palmer, 1994).

Models can be trained to learn how humans code (Rietz et al., 2020). Human–AI collaboration frameworks for TA have been proposed (Gauthier and Wallace, 2022; Gebreegziabher et al., 2023; Jiang et al., 2021).

QR and NLP researchers are now exploring the opportunity to leverage LLMs for TA. Xiao et al. (2023) investigate the use of LLMs to support deductive coding: they combine GPT-3 with an expert-drafted codebook. Gao et al. (2023) propose a user-friendly interface for collaborative qualitative analysis, leveraging LLMs for initial code generation and to help in decision-making. Paoli (2023) studies the potential usage of ChatGPT for TA following the six TA phases proposed by Braun and Clarke (2006). In the theme refinement phase, the author generates three versions of themes by modifying the temperature parameter,[4] after which the themes are reviewed by a HC (the author).

Coding can be divided into inductive coding and deductive coding. Inductive coding is a bottom-up process of developing a codebook, and deductive coding is a top-down process in which all data

---

[4]The temperature parameter determines the output randomness: higher values yield more random output. https://platform.openai.com/docs/api-reference/chat/create

Figure 2: An exemplar for initial code extraction. Green depicts the quoted sentence, blue indicates the definition of a code, and red represents the code.

is coded using the codebook. We complete both inductive and deductive coding via human–AI collaboration and showcase the feasibility of our approach using two existing open-access datasets.

## 3 Human–LLM Collaboration Framework

Figure 1 illustrates the framework, which consists of four steps.

**Data Familiarization and Exemplar Generation** We used in-context learning (ICL) to design the paradigm of prompt for the initial code extraction (Garg et al., 2022). To write the exemplars for initial code extraction, the HC must be familiar with the qualitative data. Each HC produces four to eight exemplars, following Min et al. (2022). Each exemplar includes a response for an open-ended question and the associated actions. Given a free-text response may contain multiple codes, we think "step by step" and "code by code" that were inspired by chain-of thought (CoT) prompting (Wei et al., 2022; Kojima et al., 2022). To code a response, the format of an action is "quote refers to definition of the code. Thus, we got a code: code". Figure 2 shows one instance. The quote is the sentence in the response that is related to the code.

**Initial Code Generation** At this step, codes are extracted that capture the semantic and latent meanings of the free-text responses at a fine granularity. The core of the prompt is the exemplar generated in the first step. The prompt also contains the task goal and the open-ended question, which provide the MC with a clear understanding of the purpose of the study. The prompts we used are shown in Appendix D.

**Code Refinement and Initial Codebook Generation** The model groups the similar initial codes into certain codes which the MC and HC begin to refine. Step 3 in Figure 1 illustrates a cycle of the discussion process. First an HC reviews the codes and edits the codes if necessary. If the HC modifies any code, he/she states the changes and provides the rationales behind them. After evaluating the changes and the rationales, the MC indicates agreement or disagreement with explanations. The HC then reviews the MC's responses and revises the code if needed. The HC initiates a new cycle by editing the codes and providing justification. Such refinement continues until the HC and MC are both satisfied with the result. The refined codes are then used to generate the initial codebook.

**Theme Identification and Evaluation** The MC and HC generate the final codebook by identifying the themes of the codes. MC and HC code the whole qualitative data according to the final codebook. Cohen's $\kappa$ agreement (Cohen, 1960) is calculated to evaluate the work quality of the MC and HC.

## 4 Evaluation

### 4.1 Metrics

The inner-annotator agreement (IAA) is the main method for evaluating the quality of coding (Krippendorff, 2011). IAA measures the clarity and interpretability of a codebook among the coders. A higher IAA suggests the codebook is clear and effectively captures the meaning of the data. We use Cohen's $\kappa$ to compare the results of the two datasets. In addition, to evaluate whether the codes from our framework meet the research goal of the original work, we treat the codes generated by the authors of the two datasets as the gold label. We calculate the cosine similarity between codes developed by us and those proposed by the original authors. We used the text-embedding-ada-002 embedding provided by OpenAI to embed the codes and then calculated their cosine similarity.

### 4.2 Setting

**MS** In our case study, we focused on the first-asked question because the authors mentioned that some participants provided the same or similar answers to all four questions. The sampled question is "What's the first thing that comes into your mind about this track?" Preprocessing yielded 35 responses for our study.

| | Music Shuffle | Password Manager | | |
|---|---|---|---|---|
| | All | Seen | Unseen | All |
| HC + MC | **0.87** | **0.8** | **0.82** | **0.81** |
| Coder3 + MC | 0.54 | 0.47 | 0.34 | 0.47 |
| Coder3 + HC | 0.62 | 0.48 | 0.38 | 0.43 |
| Gold | 0.66 | – | – | 0.77 |

Table 1: Cohen's $\kappa$ of the two cases. The HC, MC, and Coder 3 used the same codebook for final coding. `Seen` and `Unseen` indicate the data was sampled from the responses used or not used for developing the codebook, where `All` means `Seen + Unseen`.

| | MS | PM |
|---|---|---|
| Similarity | 0.8864 | 0.8895 |
| Accuracy | 0.84 | 0.83 |
| Recall | 0.72 | 0.87 |

Table 2: Cosine similarity, accuracy, and recall between the codes extracted by our framework and the dataset's authors.

**PM** Of the eight open-ended questions, we selected the first question—"Please describe how you manage your passwords across accounts"—as this question is general, and seems to have elicited better responses. Due to the GPT input limitations, we could not feed all the data into GPT-3.5. Thus we randomly sampled 100 responses for codebook development. Such leveraging of partial responses for codebook development has been used by Sanfilippo et al. (2020). We used the 100 responses for codebook development and in the evaluation phase we sampled 20 samples from the two data pools for labeling and IAA calculation, respectively.

Following the MS and PM workflow, we developed a codebook using our framework with a HC and the MC for each case. The HC and MC coded all the responses. We invited another coder (Coder 3) to code the responses using both codebooks developed by the HC and MC. The detailed information of the datasets is in Appendix C.

### 4.3 Results and Discussion

The results are presented in Table 1: HC + MC indicate Cohen's $\kappa$ of HC and MC annotated data, and Coder 3 + MC and Coder 3 + HC represent Cohen's $\kappa$ of Coder 3 and MC labeled data and Coder 3 and HC labeled data, respectively. We treat the result of the original studies as the gold standard for reference. Table 2 shows the result of cosine similarity, accuracy, and recall.

**LLM is suited for thematic analysis.** The results of the MS case study suggest that HC+MC achieves the best agreement compared to all other settings, where the $\kappa$ of HC+MC is 0.87 (almost perfect agreement), and that of Gold is 0.66 (substantial agreement). Further, even though the $\kappa$ decreases in the Coder 3 settings, Coder 3+MC and Coder 3+HC still exhibit similar agreement compared to Gold: The $\kappa$ of Coder 3+MC and Coder 3+HC are 0.54 (moderate to substantial agreement) and 0.62 (substantial agreement), respectively. This indicates that our framework achieves reasonable performance. In the results of the PM case study, HC+MC ($\kappa = 0.81$: almost perfect agreement) outperforms Gold ($\kappa = 0.77$: substantial agreement). The cosine similarity also shows that the MC-extracted codes capture the semantic meaning of the codes provided by the authors of the datasets. We thus conclude that the human–LLM collaboration framework can perform as well as the two human coders on thematic analysis, but with one human coder instead of two, which reduces labor and time demands.

**A discrepancy discussion process might be necessary.** The relatively poorer results of Coder 3 still indicate a nearly fair to moderate agreement. The $\kappa$ of Coder 3+MC and Coder 3+HC are (0.47, 0.48), (0.34, 0.38), and (0.47, 0.43) for the three conditions (i.e., Seen, Unseen, and All) , respectively. The PM authors mentioned that if the agreement is lower than $\kappa = 0.7$, they discussed the result and started a new coding round to re-code the data. They indicated an average 1.5 rounds of re-coding. Similarly, our results indicate a need for such a mechanism. We leave this as future work.

**Developing the codebook using partial data is a feasible approach.** In the Unseen condition, the HC+MC agreement is high ($\kappa = 0.82$: almost perfect agreement). Additionally, the cosine similarity between the codes extracted by MC and the gold codes provided by the authors of the dataset is 0.8895, which indicates the codes effectively capture the semantic meaning of the gold codes. These results imply that using partial data for codebook development is a viable way to address LLM input size limitations for TA.

### 4.4 Error Analysis

To investigate why the IAA dropped significantly in the results of Coder 3 with MC and HC, we pull

|          | MS | PM |
|----------|----|----|
| Ambiguity | 5  | 9  |
| Granularity | 8 | 7 |
| Distinction | 9 | 14 |
| **Total** | 22 | 30 |

Table 3: The number of mismatched responses in each category in the two different datasets.

|           | MS (All) | PM (Seen) | PM (Uneen) | PM (All) |
|-----------|----------|-----------|------------|----------|
| MC+Coder3 | 0.54/0.56 | 0.47/0.72 | 0.34/0.50 | 0.47/0.62 |
| HC+Coder3 | 0.62/0.72 | 0.48/0.69 | 0.38/0.54 | 0.43/0.62 |

Table 4: The number before "/" is the IAA calculated by code, and the number after "/" represents the IAA calculated by theme.

the sample that they coded differently and the codebook. We found three major reasons: **Ambiguity**, **Granularity**, and **Distinction** that led the low IAA of Coder 3 with MC and HC. Table 3 is the analysis of the samples that the three coders (MC, HC, Coder 3) coded differently.

**Ambiguity**    Ambiguity indicates that the coders coded the different codes that actually belong to the same theme. For instance, "Digital Notes" and "Notes" are both under the theme "Written Records and Notes." Given a free-text response, "*Keep them in a notes tab on my phone*", the coders might code this response as "Digital Notes" or "Notes." These two codes belong to the same theme and are correct but would drop the IAA if one coder selects "Digital notes" and the other selects "Notes." We found that among 22 mismatched responses 5 of them in MS dataset were due to the ambiguity of the codes. In PM dataset, 9 out of 30 are categorized as the reason. To further confirm if the ambiguity of the codes influences the IAA, we calculated the IAA based on the theme. Table 4 represents the result. The result suggests that all the IAA improved, which indicates that the ambiguity of the codes is one factor leading to the low IAA.

**Granularity**    Granularity denotes the granularity of the coding work. We found that HC and MC coded more finely than the Coder 3 in 8 out of 22 and 7 out of 30 mismatched responses in MS and PM, respectively. For instance, for the response, "*Sleeping, calm I saved it because I liked the way it sounded, I thought it would be a good song to fall asleep to. I enjoy listening to this song because it sounds sweet, like a cute sweet not cool sweet.*" Both HC and MC coded this response as "Positive", "Relaxing", and "sleep-inducing", where Coder 3

missed the "Positive".

**Distinction**    The other mismatched responses are due to the coders assigning different codes for a given response. This could be a result of their varied familiarity with the responses and the codebook, leading them to interpret the response or code differently. For instance, Coder 3 stated that sometimes it is hard to find a code that fits the response. As a result, Coder 3 can only identify the most appropriate code for the response.

Overall, we identified the three main factors that led the IAA drop significantly for Coder 3 with MC and HC. However, we argue that thematic analysis is inherently subjective and biased since the coders must align to the research questions when making the codebooks (Braun and Clarke, 2006; Vaismoradi et al., 2016). They generated the codebook based on their expertise in the topic of the responses and the research goal. Moreover, such subjective nature is not specific to our work. Instead, it is common for any thematic analysis. It is arguable whether bias is a drawback for thematic analysis since the goal is to provide a way to understand the collected free-text data. For instance, even though the codebooks in our study are biased, they have captured the semantic meaning of the data to the research questions, and the cosine similarity between our codebooks and the codebooks from the dataset is high (0.8864 and 0.8895). The iterative nature of codebook generation leads the codebook to be less biased. As we mentioned in the discussion section, a discrepancy discussion process might be a solution to improve the IAA.

## 5   Conclusion

We propose a framework for human–LLM collaboration (i.e., LLM-in-the-loop) for thematic analysis (TA). The result of two case studies suggests that our proposed collaboration framework performs well for TA. The human coder and machine coder exhibit almost perfect agreement ($\kappa = 0.87$ in Music Shuffle and $\kappa = 0.81$ in Password Manager), where the agreement reported by the dataset authors are $\kappa = 0.66$ for Music Shuffle and $\kappa = 0.77$ for Password Manager. We also address the LLM input size limitation by using partial data for codebook development, with results that suggest this is a feasible approach.

## Limitations

There are four main limitations in this work.

**Prompts**  While we designed specific prompts for each step that successfully elicited the desired output to achieve our goals, it is important to note that we cannot claim these prompts to be the most optimal or correct ones. As mentioned by Wei et al. (2022), there is still rooms to explore the potentiality of prompt to elicit better outputs from LLMs. We believe that there might be better versions of prompts that can achieve better outcomes.

**Application**  The survey or interview data might not be able to be put into a third-party platform like OpenAI. Specifically, certain Institutional review boards (IRB) may have regulations that the data should be stored on the institution's server. This is especially relevant for data that involves sensitive information, such as health-related studies. Therefore, the users have to check the IRB document or any regulations before they use our method for thematic analysis.

**Model**  We only studied the cases on GPT-3.5, while many off-shelf LLMs are available [5]. Therefore, we can not guarantee using other LLMs can achieve the comparable result. However, we have demonstrated the effectiveness of our proposed framework with GPT-3.5. Future work could adopt this framework to other Open-sources LLMs, such as Falcon (Almazrouei et al., 2023) and LLaMA (Touvron et al., 2023). However, the computational resources and cost might be the potential limitations.

**Data**  The dataset, Music Shuffle, was published in 2020, while the training data for GPT-3.5 extends until September 2021. Therefore, GPT might have encountered or been trained on a similar dataset before. However, as most of the data are regulated by the IRB and the privacy concerns, it is challenging to find a dataset containing the raw responses data and codebook for reuse. Therefore, we still decided to select this dataset for our study. The other dataset, Password Manager, was published in 2022, which does not have the concern, and our proposed method also performs well on this dataset.

## Ethics Statement

The datasets we used in this work are all public for reuse. All the data do not contain privacy-sensitive information. Therefore, we used these two datasets with OpenAI GPT3.5 model. However, the other

---

[5]https://github.com/Hannibal046/Awesome-LLM

users need to confirm with their IRB before they used the proposed method for thematic analysis.

## Acknowledgements

We thank Katie Zhang and Zekun Cai for helping us with the thematic coding tasks. We thank Yi-Li Hsu for fruitful discussions. Thanks as well to the anonymous reviewers for their helpful suggestions. The works of Aiping Xiong were in part supported by NSF awards #1915801, #1931441 and #2121097. This work is supported by the National Science and Technology Council of Taiwan under grants 111-2634-F-002-022- and 112-2222-E-001-004-MY2.

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

## A Qualitative Research and Thematic Analysis

Qualitative research, which uses techniques such as interviews and open-ended survey questions to collect free-text responses (Wertz, 2011; Richards and Hemphill, 2018), has been widely used to identify patterns of meanings and gain insight from participants via analysis of qualitative data. One common method for analyzing qualitative data is thematic analysis (Braun and Clarke, 2006, 2012, 2019). For reliable coding, at least two human coders must 1) become familiar with the data, 2) generate a codebook, and 3) iteratively code the data until an acceptable inter-rater agreement level is reached. Thus TA is labor-intensive and time-consuming.

## B Model and Parameters

We used OpenAI's `gpt-3.5-turbo-16k` LLM as GPT outperforms other LLMs on various NLP tasks (Liang et al., 2022). Second, although GPT-4 is smarter than and superior to GPT-3.5, and the input context size of GPT-4 (32,768 for `gpt-4-32k`) is longer than that for GPT-3.5 (16,384 for `gpt-3.5-turbo-16k`), there are more constraints on GPT-4.[6] For instance, the tokens per minute (TPM) limit for GPT-4 is 40,000, whereas that for GPT-3.5 is 180,000 (`gpt-3.5-turbo-16k`). Also, as GPT-4 is still in beta, not everyone can access it. Finally, GPT-3.5 remains a powerful LLM compared to others, and it is cheaper.

We follow OpenAI's default parameter settings except the temperature, which we set to $0$ to ensure our study is reproducible. $0$ indicates low output randomness: hence the model is less creative.

## C Datasets

**Music Shuffle (MS)** This dataset facilitates research on salient aspects when people listen to music on their personal devices. The survey includes four open-ended questions, and comprises 397 participants. The authors combined all the responses of the four questions for TA.

**Password Manager (PM)** This is a record of password manager usage and general password habits at George Washington University (GWU). The survey includes eight open-ended questions, comprising 277 participants from GWU's faculty, staff, and student body. In the original study, one coder coded all the responses and developed the codebook. Later, another coder coded $20\%$ of the responses and calculated Cohen's $\kappa$. Our selected survey questions has 280 responses ($n = 280$).

---

[6]https://platform.openai.com/docs/guides/rate-limits/overview

# D  Prompts

---

**Prompt: Initial Codes Generation**

**Task:** {The goal of the study}

Here are examples for how to generate the codes. For each example, you will see one response with the codes step by step.

These responses are the answer of the question: {Survey Question}.

Each generated code have the format: ' quote ' refers to /mentions ' definition of the code '. Therefore, we got a code: ' Code '.

Exemplars: {4 to 8 exemplars}

---

Prompt 1: The prompt for initial codes generation.

---

**Prompt: Code Grouping**

Here is the survey question: {Survey Question}

{Initial Codes}

Please organize the codes into themes in JSON format. Ensure that each code belongs to only one theme. Assign a name to each theme.

If there are any duplicate codes, please merge them into a single entry.

The expected output format should follow this structure: <Name of the theme>: <List of codes and their definition belonging to the theme>

---

Prompt 2: The prompt for grouping the codes.

---

**Prompt: Code Refinement**

Here is your version.
{Themes proposed by the Machine Coder }
Here is the revised version.
{Revised Themes by the Human Coder }
{Actions and Reasons by the Human Coder }

What do you think?
Please generate the revised themes.
Please list the parts with which you agree and disagree and the reason in JSON.

---

Prompt 3: The prompt for discussion.