# OpenReview forum: "LLM-in-the-loop: Leveraging Large Language Model for Thematic Analysis"
_EMNLP/2023/Conference — EMNLP 2023 Findings_

### Official Review · Reviewer_WzFN · 2023-08-03

**Soundness:** 3

**Excitement:**

2: Mediocre: This paper makes marginal contributions (vs non-contemporaneous work), so I would rather not see it in the conference.

**Missing References:**

Why conduct Initial Code Generation instead of using unsupervised topic categorization to identify the themes?

Lennon et al., 2021 generates unsupervised topic categories (without LDA, which was mentioned in the submission) to perform thematic analysis:
Lennon, Robert P., et al. "Developing and testing an automated qualitative assistant (AQUA) to support qualitative analysis." Family Medicine and Community Health 9.Suppl 1 (2021).

**Paper Topic And Main Contributions:**

The paper describes how a human can use prompts to train an LLM to  generate “codes” for response text to questions (e.g., such as from a semi-structured interview as used in Social Science), use prompts to create themes from the identified codes, and then prompt the LLM to code the dataset using the identified codes. Thus, the LLM serves as a “machine coder” that is taught by a “human coder”. The research question was whether the coding quality of the LLM is comparable to a human. The paper compared agreement between the LLM and human coder in their generated codes and found good agreement. However, the agreement with a third coder was not as good.

**Reasons To Accept:**

Paper provides a framework for using prompts to teach LLMs to code free-text answers where the labels used are not already known or pre-defined.

Paper tested on two datasets and found good agreement in coding between the human who prompted the LLM to perform the coding.

**Reasons To Reject:**

The paper may be of more interest in a venue where many participants require coding of responses to semi-structured interviews.

The paper topic is not about empirical methods for NLP. Rather, it describes an application of how prompting an LLM can be used for a task that is common in disciplines where structured interviews need to be analyzed and the set of labels to be used for labeling the data needs to be determined.

For evaluating generalization, I would like to see several more coders in the evaluation, more than the current one who worked with the LLM and one outside coder. Cohen’s kappa for Coder3 was much lower when matched to the LLM or human. One interpretation is that the LLM did a good job learning the codes from the human that trained it, but those codes have discrepancies with Coder3’s interpretation. Perhaps a more appropriate framework would be for humans to define the set of codes to use (possibly using techniques that have been proposed for helping coders to define the codes) and then training a model that included an LLM to perform the coding. That is, have the “discrepancy discussion” (lines 269-280) before using an LLM.

The evaluation section needs to better describe the dataset characteristics to help understand the difficulty of the task. Although Section 4.2 Setting states the one question used from each dataset that is coded, a description of each dataset is needed.

**Reproducibility:**

3: Could reproduce the results with some difficulty. The settings of parameters are underspecified or subjectively determined; the training/evaluation data are not widely available.

**Reviewer Confidence:**

4: Quite sure. I tried to check the important points carefully. It's unlikely, though conceivable, that I missed something that should affect my ratings.

**Typos Grammar Style And Presentation Improvements:**

The section 4. Evaluation seems to be incomplete and is hard to read. It may have been cut down from a longer version, since many items are referred to before they are defined. For example:
Line 200: “the datasets”, no datasets were mentioned earlier, although in lines 133 and 198  they state “our approach using two existing open-access datasets.”
At the end of Section 4.2, it’s stated that the “detailed information of the datasets is in Appendix A.4”. A brief description of the of the datasets should be given in the paper itself. How many responses were there to the two selected questions? How many codes did the authors of the dataset define? Was that number known to the coders or the LLM? If the number of codes was unknown, how was the comparison to the “gold standard” performed?

Table 1 results would be clearer if the “Gold” label was replaced by “MC + Gold” to indicate what was compared. I’d also like to see results for “HC + Gold” and “Coder3 + Gold” to see the agreement between the Gold codes against each of the human codes, in addition to against the LLM codes.

In lines 191-195, Krippendorff is mentioned in the context of inter-annotator agreement, but then it’s stated that Cohen’s kappa was used to compare the results of two coders. Why not use Krippendorff’s alpha?

How are text passages handled when more than one code should be assigned to the passage?

---

> ### Author Rebuttal · Authors · 2023-08-29
>
> Thanks for your constructive review. In the following, your concerns are addressed point-by-point.
>
> ### Weakness 1,2 NLP Venue
> This work is an interdisciplinary research, which involves qualitative research and human-centered NLP. We proposed a human-LLM collaborative framework for thematic analysis (TA). TA involves NLP techniques, such as extracting codes from the responses, grouping the codes into themes, and labeling the responses with codes. Further, LLM recently attracted peak attention in the NLP community. For instance, the theme track of EMNLP this year is Large Language Models and the Future of NLP. As a result, we believe that it is appropriate to submit this work to EMNLP.
>
> ### Weakness 3 Evaluation
> We evaluated the effectiveness of the idea by studying two different datasets with three coders: MC, HC, and coder 3. The good agreement can prove the idea is actionable. It is for sure nice to have more coders. However, we only evaluated with three coders due to limited resources and pages. We believe that the results from the two different datasets can demonstrate the effectiveness of the framework.
>
> ### Weakness 4 Dataset Description
> We don't have enough space to put detailed information about the dataset due to the page limit. We will add more descriptions about the dataset to address this concern in our next version.
>
> ### Missing Reference
> We conducted initial code generation because we followed the thematic analysis procedure, as we mentioned in the paper. Further, we aim to extract the codes that can answer the research question of the survey, and thus, we did not simply use an unsupervised topic modeling method to categorize the codes. Even though an unsupervised topic modeling approach can extract topic themes that represent the responses, these responses may not be able to answer the research question. We can expand our related work to include this aspect when space allows.
>
> ### Presentation- Gold Label
> As we mentioned in Sec. 4.1 (Line 197-200), the "Gold" indicated the IAA reported by the authors of the dataset. The authors of the two datasets coded the responses based on their developed codebook. On the other hand, we coded the responses based on the codebook developed by MC and HC. As a result, we cannot calculate the agreement between the gold standard (coded by the authors of the dataset) and ours (HC, MC, and coder 3).
>
> ### Presentation- Cohen's Kappa
> The authors of the two datasets reported Cohen's Kappa. To compare the result, we calculated Cohen's Kappa as the Inner-Annotator-Agreement.
>
> ### Presentation- More than one code
> Following the same procedure as the authors of the dataset, one response can have more than one code.
>
> Thanks again for your helpful suggestions. We will revise the paper based on your comments and add the codebook qualitative analysis and discussion to the revised version.

---

### Official Review · Reviewer_Mnmq · 2023-08-06

**Soundness:** 3

**Excitement:**

3: Ambivalent: It has merits (e.g., it reports state-of-the-art results, the idea is nice), but there are key weaknesses (e.g., it describes incremental work), and it can significantly benefit from another round of revision. However, I won't object to accepting it if my co-reviewers champion it.

**Paper Topic And Main Contributions:**

The paper proposes A human-in-the-loop framework for thematic Analysis (TA). Traditionally, TA has been done by 2 humans. The paper proposes replacing one of the human's with ChatGPT. The framework consists of 4 steps : Data familiarization, initial code generation, refinement and final coding. The human coder + machine coder team gets higher Inter-annotator-agreement than the gold standard on 2 datasets.

**Questions For The Authors:**

1) What happens if both coders are machines?

**Reasons To Accept:**

1) The human coder + machine coder has higher IAA than the gold standard.
2) The framework facilitates discussesion between a human and LLM, which can reduce errors caused by hallucination.

**Reasons To Reject:**

1) Some concepts like 'QR', 'code' are not clearly explained to the reader.
2) only one out of the 8 questions is studied on the password management data.
3)  cosine similarity between generated codes and gold standard codes is reported. However, it is not a sufficient metric. Other metrics like accuracy, recall should be reported.
4) The analysis of the extracted codes and the missed codes is not provided.

**Reproducibility:**

3: Could reproduce the results with some difficulty. The settings of parameters are underspecified or subjectively determined; the training/evaluation data are not widely available.

**Reviewer Confidence:**

3: Pretty sure, but there's a chance I missed something. Although I have a good feel for this area in general, I did not carefully check the paper's details, e.g., the math, experimental design, or novelty.

**Typos Grammar Style And Presentation Improvements:**

Full form of QR should be provided.
meaning of 'code' and 'codebook' should be made clear early in the paper.
Captions of figures should be strengthened

---

> ### Author Rebuttal · Authors · 2023-08-29
>
> Thanks for your constructive review. In the following, your concerns are addressed point-by-point.
>
> ### Weakness 2
> We evaluated the effectiveness of the idea by studying two different datasets. The result, Cohen's Kappa, for the two datasets, 0.87 and 0.81, respectively, can prove the idea is actionable. Moreover, we believe that the results from the two different datasets can demonstrate the effectiveness of the framework better than results from one dataset but many similar questions. That is, it is nice to have more experiments. However, all eight questions in PM are of the same type, i.e., qualitative responses to open-ended questions. Further, we selected that specific question to study for two main reasons. First, as we mentioned in the paper, that question is more general (*"Please describe how you manage your passwords across accounts "*) compared to other questions, such as *"Why did you use that creation strategy for your most recent GW account password?"* Second, the dataset's author stated that the response quality of the first question is the best, and the question we studied is the first question of the dataset. As a result, the responses to the question we selected to study in PM have the best quality compared to other questions. Due to limited resources and pages, we only selected one out of eight questions in PM.
>
> ### Weakness 3
>
> |  | Music Shuffle | Password Manager |
> | -------- | -------- | -------- |
> | Accuracy    | 0.84 | 0.83 |
> | Recall | 0.72 | 0.87 |
>
> The table indicates the accuracy and recall of the codes extracted by our framework and the codes extracted by the dataset’s authors. We will add them to the revised version.
>
> ### Weakness 4
> We conduct an analysis on the responses that machine coder (MC), human coder (HC) and Coder 3 coded differently.
>
>
>
> |           | Music Shuffle (MS) | Password Manager (PM) |
> | --------  | -------- | -------- |
> | Ambiguity | 5 | 9 |
> | Granularity | 8 | 7 |
> | Distinction | 9 | 14 |
> | **Total** | **22** | **30** |
>
> **Table1.** Analysis of the mismatched responses
>
>
>
> | | MS (All) | PM(Seen) | PM(Unseen) | PM(All) |
> | -------- | -------- | -------- |-| - |
> | MC+Coder 3 | 0.54/0.56 | 0.47/0.72 | 0.34/0.50 | 0.47/0.62 |
> | HC+Coder 3 | 0.62/0.72 | 0.48/0.69 | 0.38/0.54 | 0.43/0.62 |
>
> **Table2.** The former number is the IAA calculated by code (Level 3), and the letter represents the IAA calculated by theme (Level 2).
>
> To investigate why the inner-annotator agreement (IAA) dropped significantly in the result coder 3 with machine coder (MC) and human coder (HC), we pull the sample that they coded differently and the codebook. We found three major reasons: **Ambiguity**, **Granularity**, and **Distinction** that lead the low IAA of coder 3 with MC and HC. Table 1 is the analysis of the samples that the three coders (MC, HC, Coder3) coded differently.
>
> **Ambiguity:**
> Ambiguity indicates that the coders coded the different codes (Level 3) that actually belong to the same theme (Level 2). For instance, “Digital Notes” and “Notes” are both under the theme “Written Records and Notes”. Given a free-text response, “Keep them in a notes tab on my phone”, the coders might code this response as “Digital Notes” or “Notes”. These two codes belong to the same theme and are correct but would drop the IAA if one coder selects “Digital notes” and the other selects “Notes”. We found that among 22 mismatched responses 5 of them in the Music Shuffle (MS) dataset were due to the ambiguity of the codes. In the Password Manager (PM) dataset, 9 out of 30 are categorized as the reason. To further confirm if the ambiguity of the codes (Level 3) influences the IAA, we calculated the IAA based on the theme (Level 2 codes). Table 2 represents the result. The result suggests that all the IAA improved, which indicates that the ambiguity of the codes is one factor leading to the low IAA.
>
> **Granularity:**
> Granularity denotes the granularity of the coding work. We found that HC and MC coded more finely than the coder 3 in 8 out of 22 and 7 out of 30 mismatched responses in MS and PM, respectively. For instance, for the response, "Sleeping, calm I saved it because I liked the way it sounded, I thought it would be a good song to fall asleep to. I enjoy listening to this song because it sounds sweet, like a cute sweet not cool sweet." Both HC and MC coded this response as “Positive”, “Relaxing”, and “sleep-inducing”, where coder 3 missed the “Positive”.
>
> **Distinction:**
> The other mismatched responses are due to the coders assigning different codes for a given response. This could be a result of their varied familiarity with the responses and the codebook, leading them to interpret the response or code differently. For instance, coder 3 stated that sometimes it is hard to find a code that fits the response. As a result, coder 3 can only identify the most appropriate code for the response.
>
> Overall, we identified the three main factors that led the IAA drop significantly for coder 3 with MC and HC. However, we argued that thematic analysis is inherently subjective and biased since the coders must align to the research questions when making the codebooks [1,2]. They generated the codebook based on their expertise in the topic of the responses and the research goal. Moreover, **such subjective nature is not specific to our work. Instead, it is common for any thematic analysis.** It is arguable whether bias is a drawback for thematic analysis since the goal is to provide a way to understand the collected free-text data. For instance, even though the codebooks in our study are biased, they capture the semantic meaning of the data to the research questions, and the cosine similarity between our codebooks and the codebooks from the dataset is high (0.8864 and 0.8895). The iterative nature of codebook generation leads the codebook to be less biased.  As we mentioned in the paper, a discrepancy discussion process might be a solution to improve the IAA (lines 269-280).
>
> [1] Braun, V., & Clarke, V. (2006). Using thematic analysis in psychology. Qualitative research in psychology, 3(2), 77-101.
> [2] Vaismoradi, M., Jones, J., Turunen, H., & Snelgrove, S. (2016). Theme development in qualitative content analysis and thematic analysis.
>
>
> ### Question 1
> In this work, we focus on the idea of a human-LLM collaboration framework for thematic analysis. We also believe that leveraging two MC, i.e., two different LLMs, is a valuable direction to explore. We leave this as the future direction.
>
> ### Typo, Presentation, and Weakness 1
> We will address the presentation, such as QR, code, and codebook, in the revised version.
>
> Thanks again for your helpful suggestions. We will revise the paper based on your comments and add the codebook qualitative analysis and discussion to the revised version.

---

### Official Review · Reviewer_rocS · 2023-08-10

**Typos Grammar Style And Presentation Improvements:** 1. Minor - Presentation of Table 1 sh…
**Soundness:** 4

**Excitement:**

4: Strong: This paper deepens the understanding of some phenomenon or lowers the barriers to an existing research direction.

**Paper Topic And Main Contributions:**

This paper performs an exploratory study on the effectiveness of LLMs in performing thematic analysis on qualitative data. They aim to alleviate the labor-intensive and time-consuming nature of qualitative analysis by introducing their LLM-based framework to work in conjunction with humans to extract a meaningful understanding of qualitative data.

**Questions For The Authors:**

A. As mentioned above, a qualitative analysis of codebook generation and coding of free-text responses using this approach would strengthen the paper. Is there such an example you can provide?
B. Have you considered comparing your LLM-based framework to other language model-based methods (fine-tuned BERT for example)

**Reasons To Accept:**

1. Strong problem motivation and problem formulation in the context of recent state-of-the-art literature.
2. Clear and concise presentation of the problem and solution
3. Work is novel, and the application of this framework has significant downstream potential for qualitative research applications.
4. Again, the idea and domain are very novel, the results and the analysis provide a good addition to the LLM theme.
5. Result is reproducible easily, and seems to be good.

**Reasons To Reject:**

1. Minor - I think a much more comprehensive and data-intensive analysis would improve this paper significantly but since it is a short paper this isn't a strong negative against what has been done by the authors.
2. I am unsure about the technical novelty of the approach - the paper appears to be simply doing prompt engineering on ChatGPT to meet an end goal. I like the domain and the framework but the experimental results, while positive, show the strength of ChatGPT (LLMs) in performing qualitative analysis but not the author's claim of reducing human burdens of performing TA.
3. Minor - Need to have at least one qualitative analysis of how the machine coder and human coder are selecting similar or different "codes" needed for completeness.

**Reproducibility:**

5: Could easily reproduce the results.

**Reviewer Confidence:**

4: Quite sure. I tried to check the important points carefully. It's unlikely, though conceivable, that I missed something that should affect my ratings.

---

> ### Author Rebuttal · Authors · 2023-08-29
>
> Thank you for your valuable suggestions for our work. We are delighted that you acknowledge the contributions of our work to qualitative research.
>
> ### Weakness 1, 3, and Question A
> We conduct an analysis on the responses that machine coder (MC), human coder (HC) and Coder 3 coded differently.
>
>
>
> |           | Music Shuffle (MS) | Password Manager (PM) |
> | --------  | -------- | -------- |
> | Ambiguity | 5 | 9 |
> | Granularity | 8 | 7 |
> | Distinction | 9 | 14 |
> | **Total** | **22** | **30** |
>
> **Table1.** Analysis of the mismatched responses
>
>
>
> | | MS (All) | PM(Seen) | PM(Unseen) | PM(All) |
> | -------- | -------- | -------- |-| - |
> | MC+Coder 3 | 0.54/0.56 | 0.47/0.72 | 0.34/0.50 | 0.47/0.62 |
> | HC+Coder 3 | 0.62/0.72 | 0.48/0.69 | 0.38/0.54 | 0.43/0.62 |
>
> **Table2.** The former number is the IAA calculated by code (Level 3), and the letter represents the IAA calculated by theme (Level 2).
>
> To investigate why the inner-annotator agreement (IAA) dropped significantly in the result coder 3 with machine coder (MC) and human coder (HC), we pull the sample that they coded differently and the codebook. We found three major reasons: **Ambiguity**, **Granularity**, and **Distinction** that lead the low IAA of coder 3 with MC and HC. Table 1 is the analysis of the samples that the three coders (MC, HC, Coder3) coded differently.
>
> **Ambiguity:**
> Ambiguity indicates that the coders coded the different codes (Level 3) that actually belong to the same theme (Level 2). For instance, “Digital Notes” and “Notes” are both under the theme “Written Records and Notes”. Given a free-text response, “Keep them in a notes tab on my phone”, the coders might code this response as “Digital Notes” or “Notes”. These two codes belong to the same theme and are correct but would drop the IAA if one coder selects “Digital notes” and the other selects “Notes”. We found that among 22 mismatched responses 5 of them in the Music Shuffle (MS) dataset were due to the ambiguity of the codes. In the Password Manager (PM) dataset, 9 out of 30 are categorized as the reason. To further confirm if the ambiguity of the codes (Level 3) influences the IAA, we calculated the IAA based on the theme (Level 2 codes). Table 2 represents the result. The result suggests that all the IAA improved, which indicates that the ambiguity of the codes is one factor leading to the low IAA.
>
> **Granularity:**
> Granularity denotes the granularity of the coding work. We found that HC and MC coded more finely than the coder 3 in 8 out of 22 and 7 out of 30 mismatched responses in MS and PM, respectively. For instance, for the response, "Sleeping, calm I saved it because I liked the way it sounded, I thought it would be a good song to fall asleep to. I enjoy listening to this song because it sounds sweet, like a cute sweet not cool sweet." Both HC and MC coded this response as “Positive”, “Relaxing”, and “sleep-inducing”, where coder 3 missed the “Positive”.
>
> **Distinction:**
> The other mismatched responses are due to the coders assigning different codes for a given response. This could be a result of their varied familiarity with the responses and the codebook, leading them to interpret the response or code differently. For instance, coder 3 stated that sometimes it is hard to find a code that fits the response. As a result, coder 3 can only identify the most appropriate code for the response.
>
> Overall, we identified the three main factors that led the IAA drop significantly for coder 3 with MC and HC. However, we argued that thematic analysis is inherently subjective and biased since the coders must align to the research questions when making the codebooks [1,2]. They generated the codebook based on their expertise in the topic of the responses and the research goal. Moreover, **such subjective nature is not specific to our work. Instead, it is common for any thematic analysis.** It is arguable whether bias is a drawback for thematic analysis since the goal is to provide a way to understand the collected free-text data. For instance, even though the codebooks in our study are biased, they capture the semantic meaning of the data to the research questions, and the cosine similarity between our codebooks and the codebooks from the dataset is high (0.8864 and 0.8895). The iterative nature of codebook generation leads the codebook to be less biased.  As we mentioned in the paper, a discrepancy discussion process might be a solution to improve the IAA (lines 269-280).
>
> [1] Braun, V., & Clarke, V. (2006). Using thematic analysis in psychology. Qualitative research in psychology, 3(2), 77-101.
> [2] Vaismoradi, M., Jones, J., Turunen, H., & Snelgrove, S. (2016). Theme development in qualitative content analysis and thematic analysis.
>
>
> ### Weakness 2
> In this work, our focus was on exploring the potential way to integrate humans and LLMs for conducting thematic analysis. As a result, the novelty of this study lies primarily in the framework itself.
>
> ### Question B
> We have considered adopting the same workflow on other language models, such as Llama2. We leave this as future work.
>
> Thanks again for your helpful suggestions. We will revise the paper based on your comments and add the codebook qualitative analysis and discussion to the revised version. For the presentation issues, such as the acronym QR, we will address them in the revised version.

---

### Official Review · Reviewer_Gc6Y · 2023-08-12

**Soundness:** 3

**Excitement:**

3: Ambivalent: It has merits (e.g., it reports state-of-the-art results, the idea is nice), but there are key weaknesses (e.g., it describes incremental work), and it can significantly benefit from another round of revision. However, I won't object to accepting it if my co-reviewers champion it.

**Justification For Ethical Concerns:**

No ethics concern.

**Paper Topic And Main Contributions:**

The paper explores the potential of using LLM as a collaborative agent along with a singular human agent to perform Thematic Analysis (TA) task on free-text survey responses. Previous approaches typically involve two or more human expert annotators who identify categorical themes from responses and discuss them to reach a consensus. Leveraging LLM has an obvious advantage in terms of reduction in labor and time. In the proposed approach, the human coder (HC) first identifies exemplars from the survey response data. Then, prompts are created by including the survey question, quote of the response, and identified code (along with definition or rationale) in a chain-of-thought natural language text. Guided by these prompts, the LLM or machine coder (MC) generates initial codes for the rest of the data provided to it. This is followed by a discussion phase between the HC and MC and upon agreement, a codebook is generated. This codebook is then used by HC, MC, and another human coder (Coder 3), to perform the final coding. The inter-coder reliability is evaluated using Cohen's kappa.

The work is evaluated using one LLM, specifically gpt-3.5-turbo-16k, and two public datasets, Music Shuffle (MS) and Password Manager (PM). Empirical results in Table 1, interpreted according to Landis and Koch (1977), depict an "almost perfect" (0.81-1) agreement between HC+MC in all cases (ignoring the near-miss of PM-Seen). The agreeability between Coder 3 and HC is "moderate" or "substantial" for MS and is similar to that in the Gold reference. However, the agreeability of Coder 3 with both HC and MC is categorically lower ("fair" or "moderate").

---

**[Update after rebuttal, date: 03-Sep-2023]**
Please refer to my official comment in reply to the authors' response *(no other change is being made to the text of the original review)*. As mentioned in the comment, I am increasing the excitement score but retaining the soundness score as it is.

**Questions For The Authors:**

1. In reference to Weakness #2, it would be helpful if the authors can share if any experiments were performed to validate whether the relatively lower agreeability of Coder 3 with MC and HC (compared to HC+MC) is indeed indicative of problems of biases in the codebook (biases in the codebook could still lead to "almost perfect" inter-coder reliability score for HC+MC if they would have come up during the discussion phase and affected the codebook).

**Reasons To Accept:**

1. The work **explores the potential of leveraging LLM in a collaborative approach** along with a human coder (HC) to prepare a codebook with mutual consensus that can then be used effectively by the same human coder (HC) or another human (Coder 3) to perform Thematic Analysis (TA) with decent agreeability. In the current landscape of LLM applications, this contributes insights into LLM applications for an NLP task that involves the processing and categorical analysis of free-text survey responses.

2. Empirical data shows that the codebook prepared according to the proposal is **at least effective to the extent that the human coder (HC) is in "almost perfect" agreement with the final code generated by the LLM (MC)**.

3. The use of **(a) in-context learning with exemplars in prompts, and (b) partial responses for codebook development in the PM dataset**, would provide another reference of their efficacy in the literature in relation to useful techniques for prompt engineering while using LLMs.

**Reasons To Reject:**

1. The work has only been evaluated with one LLM and two public datasets which makes it **difficult to claim that the presented results alone support the overall idea** that LLMs are effective in the Thematic Analysis (TA) task. Furthermore, the **agreeability of Coder 3 with both MC and HC is significantly lower** in the case of one of these datasets (PM), compared to the Gold reference, which further leads to the question of LLM applicability in this task remaining short of substantiated.

2. The "almost perfect" agreement between HC and MC is **perhaps indicative of the biases present in the HC to have impacted the codebook** as the LLM (MC) may have been more inclined to accept the rationale provided by the HC during the discussion phase. With two human coders, this might have led to more to-and-fro discussions to resolve differences of judgment but the other human coder is likely to have been more resilient to biases of the peer in impacting the codebook. The drop in Cohen's kappa when Coder 3 is involved also supports this possibility, and no experiments or discussion is included in the manuscript to rule this out.

3. (Relatively minor concern) In terms of a paper involving LLM, where the advantage is mainly in terms of cost (labor + time) reduction, a holistic approach is warranted where the actual costs of performing inference in terms of **carbon footprint** are also taken into account.

**Reproducibility:**

3: Could reproduce the results with some difficulty. The settings of parameters are underspecified or subjectively determined; the training/evaluation data are not widely available.

**Reviewer Confidence:**

5: Positive that my evaluation is correct. I read the paper very carefully and I am very familiar with related work.

**Typos Grammar Style And Presentation Improvements:**

Typos:
1. Line 222: "has" -> "has been"
2. Line 351: "can used" --> "used" or "can use" (both are suitable in the context of the sentence)
3. Line 354: "prposed" -> "proposed"


Presentation:
1. The cosine similarity scores and their discussion are easy to miss (lines 261-263 and 284-288). Furthermore, **I could not locate the actual score for cosine similarity for the MS dataset (lines 261-263)**.

---

> ### Author Rebuttal · Authors · 2023-08-29
>
> Thank you for your helpful and constructive review. We are glad that you find our work "potential of leveraging LLM in a collaborative approach."
>
> ### Weakness 1, 2 and Question 1
> We conduct an analysis on the responses that machine coder (MC), human coder (HC) and Coder 3 coded differently.
>
>
>
> |           | Music Shuffle (MS) | Password Manager (PM) |
> | --------  | -------- | -------- |
> | Ambiguity | 5 | 9 |
> | Granularity | 8 | 7 |
> | Distinction | 9 | 14 |
> | **Total** | **22** | **30** |
>
> **Table1.** Analysis of the mismatched responses
>
>
>
> | | MS (All) | PM(Seen) | PM(Unseen) | PM(All) |
> | -------- | -------- | -------- |-| - |
> | MC+Coder 3 | 0.54/0.56 | 0.47/0.72 | 0.34/0.50 | 0.47/0.62 |
> | HC+Coder 3 | 0.62/0.72 | 0.48/0.69 | 0.38/0.54 | 0.43/0.62 |
>
> **Table2.** The former number is the IAA calculated by code (Level 3), and the latter represents the IAA calculated by theme (Level 2).
>
> To investigate why the inner-annotator agreement (IAA) dropped significantly in the result coder 3 with machine coder (MC) and human coder (HC), we pull the sample that they coded differently to the codebook. We found three major reasons: **Ambiguity**, **Granularity**, and **Distinction** that lead the low IAA of coder 3 with MC and HC. Table 1 is the analysis of the samples that the three coders (MC, HC, Coder3) coded differently.
>
> **Ambiguity:**
> Ambiguity indicates that the coders coded the different codes (Level 3) that actually belong to the same theme (Level 2). For instance, “Digital Notes” and “Notes” are both under the theme “Written Records and Notes”. Given a free-text response, “Keep them in a notes tab on my phone”, the coders might code this response as “Digital Notes” or “Notes”. These two codes belong to the same theme and are correct but would drop the IAA if one coder selects “Digital notes” and the other selects “Notes”. We found that among 22 mismatched responses 5 of them in the Music Shuffle (MS) dataset were due to the ambiguity of the codes. In the Password Manager (PM) dataset, 9 out of 30 are categorized as the reason. To further confirm if the ambiguity of the codes (Level 3) influences the IAA, we calculated the IAA based on the theme (Level 2 codes). Table 2 represents the result. The result suggests that all the IAA improved, which indicates that the ambiguity of the codes is one factor leading to the low IAA.
>
> **Granularity:**
> Granularity denotes the granularity of the coding work. We found that HC and MC coded more finely than the coder 3 in 8 out of 22 and 7 out of 30 mismatched responses in MS and PM, respectively. For instance, for the response, "Sleeping, calm I saved it because I liked the way it sounded, I thought it would be a good song to fall asleep to. I enjoy listening to this song because it sounds sweet, like a cute sweet not cool sweet." Both HC and MC coded this response as “Positive”, “Relaxing”, and “sleep-inducing”, where coder 3 missed the “Positive”.
>
> **Distinction:**
> The other mismatched responses are due to the coders assigning different codes for a given response. This could be a result of their varied familiarity with the responses and the codebook, leading them to interpret the response or code differently. For instance, coder 3 stated that sometimes it is hard to find a code that fits the response. As a result, coder 3 can only identify the most appropriate code for the response.
>
> Overall, we identified the three main factors that led the IAA drop significantly for coder 3 with MC and HC. However, we argued that thematic analysis is inherently subjective and biased since the coders must align to the research questions when making the codebooks [1,2]. They generated the codebook based on their expertise in the topic of the responses and the research goal. Moreover, **such subjective nature is not specific to our work. Instead, it is common for any thematic analysis.** It is arguable whether bias is a drawback for thematic analysis since the goal is to provide a way to understand the collected free-text data. For instance, even though the codebooks in our study are biased, they capture the semantic meaning of the data to the research questions, and the cosine similarity between our codebooks and the codebooks from the dataset is high (0.8864 and 0.8895). The iterative nature of codebook generation leads the codebook to be less biased.  As we mentioned in the paper, a discrepancy discussion process might be a solution to improve the IAA (lines 269-280).
>
> [1] Braun, V., & Clarke, V. (2006). Using thematic analysis in psychology. Qualitative research in psychology, 3(2), 77-101.
> [2] Vaismoradi, M., Jones, J., Turunen, H., & Snelgrove, S. (2016). Theme development in qualitative content analysis and thematic analysis.
>
> ### Weakness 3
> We acknowledge the concern of carbon footprint and will add this to the limitation section.
>
> Thanks again for pointing out the concerns and suggestions. We will revise the paper based on your comments and add the codebook qualitative analysis and discussion to the revised version.

---

### Meta-Review · Area_Chair_PehC · 2023-09-15

**Recommendation:** 3

**Metareview:**

Paper Summary:
This paper proposes using large language models (LLMs) as collaborators in a human-in-the-loop framework for thematic analysis of qualitative data. A human coder provides exemplars to teach the LLM to generate codes, followed by joint codebook creation. Experiments show high agreement between human coder and LLM, but lower agreement with a separate human coder.

I will now provide a snapshot of the reviews. Four reviewers were engaged and unfortunately, I'm unable to remove one of the reviewers. I'm leaving this aspect for the SAC to evaluate and decide which one to expunge.
Summary of Reviews:

Reviewer 1 found the collaborative LLM approach novel. But only one LLM and two datasets make claims preliminary. Lower coder 3 agreement is concerning and prompts bias questions.

Reviewer 2 thought the problem motivation and presentation were strong, and called the application domain very novel. However, more comprehensive experiments are needed. Unsure of true technical novelty vs. prompt engineering.

Reviewer 3 liked the higher agreement than the gold standard, and the discussion reducing hallucination risks. But felt that some concepts were not clearly explained. The reviewer thinks that metrics like accuracy can be more evaluated in the paper and that missed code analysis is lacking.

R4 noted framework enables teaching LLMs new codes, and the human-LLM agreement was good. However, the topic may be better suited for other venues. More coders are needed in the evaluation. The reviewer also believes that the dataset currently lacks details.

In general, I find incorporating LLMs into the qualitative analysis workflow quite interesting and relevant given the application area. I think it's an eye-opener towards using LLMs to beat down annotation costs. Moreover, all four reviewers praise the novelty. However, concerns exist around evaluation methodology and broader impact. The agreement between LLM and untrained human coders is notably lower, raising questions about bias and generalization. More importantly, the majority of the reviewers recommend authors' using more datasets, and coders, and probe the transparency around disagreements. I note that the authors' rebuttal to the reviewers has shed more light on this aspect. The authors provided a more detailed analysis of the discrepancies between the LLM-trained coder and the untrained coder 3. They identify sources like code ambiguity and granularity differences that account for many of the disagreements. By calculating agreement at the theme level, they showed improved results to address reviewers' concerns. Particularly, I note that the newly adopted theme-level calculation justifies the lower coder 3 agreement as initially reported and alleviates bias concerns. The additional accuracy metrics are useful as well. Personally, I agree with the reviewers that the potential is apparent but current validation is limited and not fully answered by the reviewers and I'm hoping that the authors can improve these areas of the paper in the final draft after extending their experiments.

---

### Decision · Program_Chairs · 2023-10-07

**Decision:**

Accept-Findings

**Comment:**

Paper Summary:
This paper proposes using large language models (LLMs) as collaborators in a human-in-the-loop framework for thematic analysis of qualitative data. A human coder provides exemplars to teach the LLM to generate codes, followed by joint codebook creation. Experiments show high agreement between human coder and LLM, but lower agreement with a separate human coder.

I will now provide a snapshot of the reviews. Four reviewers were engaged and unfortunately, I'm unable to remove one of the reviewers. I'm leaving this aspect for the SAC to evaluate and decide which one to expunge.
Summary of Reviews:

Reviewer 1 found the collaborative LLM approach novel. But only one LLM and two datasets make claims preliminary. Lower coder 3 agreement is concerning and prompts bias questions.

Reviewer 2 thought the problem motivation and presentation were strong, and called the application domain very novel. However, more comprehensive experiments are needed. Unsure of true technical novelty vs. prompt engineering.

Reviewer 3 liked the higher agreement than the gold standard, and the discussion reducing hallucination risks. But felt that some concepts were not clearly explained. The reviewer thinks that metrics like accuracy can be more evaluated in the paper and that missed code analysis is lacking.

R4 noted framework enables teaching LLMs new codes, and the human-LLM agreement was good. However, the topic may be better suited for other venues. More coders are needed in the evaluation. The reviewer also believes that the dataset currently lacks details.

In general, I find incorporating LLMs into the qualitative analysis workflow quite interesting and relevant given the application area. I think it's an eye-opener towards using LLMs to beat down annotation costs. Moreover, all four reviewers praise the novelty. However, concerns exist around evaluation methodology and broader impact. The agreement between LLM and untrained human coders is notably lower, raising questions about bias and generalization. More importantly, the majority of the reviewers recommend authors' using more datasets, and coders, and probe the transparency around disagreements. I note that the authors' rebuttal to the reviewers has shed more light on this aspect. The authors provided a more detailed analysis of the discrepancies between the LLM-trained coder and the untrained coder 3. They identify sources like code ambiguity and granularity differences that account for many of the disagreements. By calculating agreement at the theme level, they showed improved results to address reviewers' concerns. Particularly, I note that the newly adopted theme-level calculation justifies the lower coder 3 agreement as initially reported and alleviates bias concerns. The additional accuracy metrics are useful as well. Personally, I agree with the reviewers that the potential is apparent but current validation is limited and not fully answered by the reviewers and I'm hoping that the authors can improve these areas of the paper in the final draft after extending their experiments.